# Hindsight Curriculum Generation Based Multi-Goal Experience Replay

## Abstract

In multi-goal tasks with sparse rewards, it is challenging to learn from tons of experiences with zero rewards. Hindsight experience replay (HER), which replays past experiences with additional heuristic goals, has shown it possible for off-policy reinforcement learning (RL) to make use of failed experiences. However, the replayed experiences may not lead to well-explored state-action pairs, especially for a pseudo goal, which instead results in a poor estimate of the value function. To tackle the problem, we propose to resample hindsight experiences based on their likelihood under the current policy and the overall distribution. Based on the hindsight strategy, we introduce a novel multi-goal experience replay method that automatically generates a training curriculum, namely Hindsight Curriculum Generation (HCG). As the range of experiences expands, the generated curriculum strikes a dynamic balance between exploiting and exploring. We implement HCG with the vanilla Deep Deterministic Policy Gradient(DDPG), and experiments on several tasks with sparse binary rewards demonstrate that HCG improves sample efficiency of the state of the art.

## 1 Introduction

Multi-goal tasks with sparse rewards present a big challenge for training a reliable RL agent. In multi-goal tasks (Plappert et al., 2018), an agent learns to achieve multiple different goals and receives no positive feedback until it reaches the position defined by the desired goal. Such a sparse rewards problem makes it difficult to reuse past experiences for that the positive feedback to reinforce policy is rare. It's impractical to carefully engineer a shaped reward function (Ng et al., 1999; Popov et al., 2017) that aligns with each task or assign a set of general auxiliary tasks (Riedmiller et al., 2018), which relies on expert knowledge. To enrich the positive feedback, Andrychowicz et al. (2017) provides HER, a novel multi-goal experience replay strategy, which enables an agent to learn from unshaped reward, e.g. a binary signal indicating successful task solving. Specifically, HER replaces the desired goals with the achieved ones then recalculates the rewards of sampled experiences. By relabeling experiences with pseudo goals, it expands experiences without further exploration and is likely to turn failed experiences into successful ones with positive feedback.

Experience relabeling makes better use of failed experiences. However, not all the achieved goals lead the origin experience to a reliable state-action visitation under the current policy. (For simplicity, we denote state-goal pairs as augmented states.) A Value function of a policy for a specific state-action pair can generalize to similar ones, in return the estimate of the value function may get worse without sufficient visitation near the state-action pair. For HER, it almost replays uniform samples of past experiences whilst goals for exploring are finite. When performing a value update, the current policy could not give a credible estimate of the universal value function (Schaul et al., 2015), if the state-action pair is not well-explored. In other words, the current policy may have difficulty in generalizing to the state-action pair. Recent works focus on improving HER by evaluating the past experiences where we sample pseudo goals from. HER with Energy-Based Prioritization (EBP)(Zhao & Tresp, 2018) defines a trajectory energy function as the sum of the transition energy of the target object over the trajectory. Curriculum-guided HER (CHER) (Fang et al., 2019b) adaptively selects the failed experiences according to the proximity to the true goals and the curiosity of exploration over diverse pseudo goals with gradually changed proportion. Though these variants select valuable goals for replay, it remains a challenge that the agent will not further explore most of the pseudo goals then it is risky to directly generalize over pseudo goals.

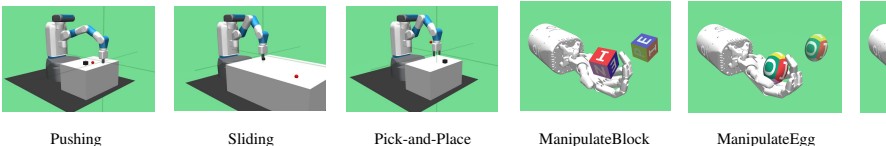

Figure 1: Tasks in the open-ended environment with sparse binary rewards. They include Pushing, Sliding and Pick-and-Place with a Fetch robotic arm as well as different in-hand object manipulations with a Shadow Dexterous Hand. The agent obtains a reward of 1 if it achieves the desired goal within some task-specific tolerance and 0 otherwise.

Humans show great capability in abstracting and generalizing knowledge but it takes a large number of experiences to learn to represent similar states with similar features. Fortunately, states with different achieved goals and desired goals may be similar intrinsically if the states and distance between them are similar. During an episode, the distance varies widely for a fixed desired goal, which has the potential for generalization. Therefore, we take advantage of relative goals-distances between the achieved goals and the desired goals-to transform data. The relative goals strategy alleviates the challenge of generalization without sufficient data. By explicitly discovering similar states in the replay buffer, it enables us to see the density of state-action visitations for unexplored goals more conveniently. The density reflects the likelihood of the corresponding state-action pair, indicating whether it is well explored. Besides, the generalization over relative goals is feasible only if the explored relative goals are widely distributed (Schaul et al., 2015). In a word, it is significant to ensure sufficient state-action visitations and maintain a balanced distribution over valid goals.

In this paper, we present to resample hindsight experiences with a relative goals strategy. The main criterion to sample experiences for a reliable generalization is based on 1) the likelihood of the corresponding state-action pair under the current policy; 2) the overall distribution of the relative goals. By constantly adjusting the distribution of goals, we propose Hindsight Curriculum Generation (HCG), which generates a replay curriculum to progressively expand the range of experiences for training. The main advantage is that it makes efficient use of hindsight experiences as well as tries to ensure the generalization over state-action pairs. From the perspective of curriculum learning, the generated curriculum can be seen as a sequence of weights on the training experiences, which guide the learning by automatically generating suitable replay goals.

Furthermore, we implemented HCG with the vanilla Deep Deterministic Policy Gradient(DDPG) (P. et al., 2015) on various Robotic Control problems. The robot, a 7-DOF Fetch Robotics arm which has a two-fingered parallel gripper or an anthropomorphic robotic hand with 24 degrees of freedom, performs the training procedure using the MuJoCo simulated physics engine (Todorov et al., 2012). During the training procedure, our method extracts and leverages information from hindsight experiences with various state-goal pairs. We experimentally demonstrated that our method improves the sample efficiency of the vanilla HER in solving multi-goal tasks with sparse rewards. Ablation studies show that our method is robust on the major hyperparameters.

## 2 BACKGROUND

In this section we briefly introduce the multi-goal RL framework, universal value function approximators and hindsight experience replay strategy used in the paper.

### 2.1 MULTI-GOAL RL

Consider an infinite-horizon discounted Markov decision process(MDP), defined by the tuple $(\mathcal{S}, \mathcal{A}, \mathcal{G}, P, r, \gamma)$, where $\mathcal{S}$ is a set of states, $\mathcal{A}$ is a set of actions, $\mathcal{G}$ is a set of goals, $P : \mathcal{S} \times \mathcal{A} \times \mathcal{S} \to \mathbb{R}$ is the transition probability distribution, $r : \mathcal{S} \times \mathcal{A} \times \mathcal{G} \to \mathbb{R}$ is the reward function, and $\gamma \in (0, 1)$ is the discount factor. In multi-goal RL, an agent interacts with its discounted MDP environment in a sequence of episodes. At the beginning of each episode, the agent receives a goal state $g \in \mathcal{G}$. In this paper we set that each $g \in \mathcal{G}$ corresponds to a goal state $s_g \in \mathcal{S}$. Moreover, we assume that given a state $s$ we can easily find a goal $g$ which is satisfied in this state. At each timestep $t$, the agent

observes a state $s_t \in \mathcal{S}$, chooses and executes an action $a_t \in \mathcal{A}$. And the agent will receive a resulting reward $r(s_t, a_t, g)$ at the next timestep $t + 1$. (For simplicity, we denote $r_t = r(s_t, a_t, g)$.) In multi-goal RL, the reward function $r$ is a binary sparse signal indicating whether the agent achieves the desired goal state:

$$r_t = \begin{cases} 1, \|\phi(s_{t+1}) - g\|_2 \leq \delta_g \\ 0, \qquad otherwise \end{cases}$$

where $\phi : \mathcal{S} \to \mathcal{G}$, a known and tractable mapping, defines the corresponding goal representation of each state, and $\delta_g$ is a task-specific tolerance threshold defined in Plappert et al. (2018).

## 2.2 UNIVERSAL VALUE FUNCTION APPROXIMATORS

Problems following the multi-goal RL framework (Plappert et al., 2018) tell RL agent what to do using an additional goal as input. The goal is fixed in each episode whilst there is more than one goal to achieve. In the continuous control problems, the agent could not afford to learn a policy for each goal. Instead, the policy should generalize not just over states but also over goals via deep neural networks. Formally, Universal Value Function Approximators (UVFA) (Schaul et al., 2015) factor observed values into separate embedding vectors for states and goals, then learn a mapping from $(s, g)$ pairs to factored embedding vectors. Let $\tau = s_1, a_1, s_2, a_2, \ldots, s_{T-1}, a_{T-1}, s_T$ denote a trajectory, which is also an episode, $R_t = \sum_{i=t}^{T} \gamma^{i-t} r_i$ denote its discounted return at every timestep $t \in [1, T]$. Let $\pi : \mathcal{S} \times \mathcal{G} \to \mathcal{A}$ denote a universal policy, $V^\pi : \mathcal{S} \times \mathcal{G} \to \mathbb{R}$ denote its value function. The objective of the agent is to learn a general value function parameterized by $\theta$ that represents the expected discounted return, i.e.

$$V^\pi(s_t, g) := \mathbb{E}\left[R_t | \theta\right],$$

or to learn a policy $\pi$ that maximizes expected discounted return. The $Q$-function $Q^\pi : \mathcal{S} \times \mathcal{G} \times \mathcal{A} \to \mathbb{R}$ also depends on goals. Notice that the transition probability distribution is independent of goals, it is possible to train an approximator to the $Q$-function using direct bootstrapping from the Bellman equation

$$Q^\pi(s_t, g, a_t) := \mathbb{E}_{s_{t+1}}\left[r_t + \gamma V^\pi(s_{t+1}, g)\right]. \tag{1}$$

## 2.3 HINDSIGHT EXPERIENCE REPLAY

Experience replay is the key strategy of off-policy RL to remember and reuse past experiences, which has shown great power in solving large sequential decision-making problems, such as Atari Games (Mnih et al., 2015) and Robotic Control(P. et al., 2015). By resampling experiences for agent training, it makes better use of experiences than on-policy RL. Let $\mu : \mathcal{S} \times \mathcal{G} \to \mathcal{A}$ denote a universal behavior policy and $\pi$ denote the target policy. In off-policy RL, the target policy can learn from experiences generated by any behavior policy as long as if $\mathbb{P}(a = \pi(s_t, g)) > 0$, we have $\mathbb{P}(a = \mu(s_t, g)) > 0$ at each $t$. If any state-action pair $(s_t, g, a)$ is unavailable for behabior policy, there will be approximation error in the estimation of $Q^\pi(s_t, g, a_t)$.

For any off-policy RL algorithm, HER modifies the desired goals $g$ in the replay transitions to some achieved goals $g'$ sampled from failed episodes. Specifically, it stores transitions not only with the original goal used for its episode but also with a subset of other goals. Notice that for UVFAs, Eq.(1) holds with relabeled experiences, which makes it possible to relabel past experiences with additional goals. HER generates additional goals using hand-crafted heuristics. It proposes various goals generation strategies, e.g. $future$ strategy that replays with $m$ random states which come from the same episode as the transition being replayed and were observed after it. The hyperparameter $m$ controls the ratio of relabeled experiences to those coming from normal experience replay.

The unexplored goal $g'$ affects the estimation of $Q^\pi(s_t, g', a_t)$, which is performed by a variant of Eq.(1)

$$Q^\pi(s_t, g', a_t) = \mathbb{E}_{s_{t+1}}\left[r_t + \gamma Q^\pi(s_{t+1}, g', a)\right], \tag{2}$$

where $a$ is sampled from $\pi$. The target policy may select an unfamiliar action $a$ at the next state-goal pair in the backed-up value estimate. More generally, the estimate, $Q^\pi(s_{t+1}, g', a)$, will be unreliable in lack of sufficient visitations near $(s_{t+1}, g', a)$. Especially when relabeling experiences without further exploration, the universal value function could not generalize well over pseudo goals.

# 3 HINDSIGHT CURRICULUM GENERATION

In this section, we introduce a hindsight experience selection method to sample valuable experiences for replay, as well as maintain the generalization of the policy. The hindsight strategy operates on preprocessed experiences with relative goals. First, we introduce the preprocessing and the generalizability assumption. Then we propose a practical clustering method to discover similar experiences and take the density of state-action visitations into account to give a more reliable value estimate. Finally, we control the range of replayed goals by maintaining a balanced distribution, which generates a hindsight curriculum to guide the learning.

## 3.1 PREPROCESSING

The previous works (Andrychowicz et al., 2017; Zhao & Tresp, 2018; Fang et al., 2019b) simply concatenate a state $s$ and a goal $g$ together as a joint input for value function, so do we. As the deep neural network has a strong generalization ability, it should learn a similar representation for similar augmented states, i.e. state-goal pairs. In particular, it relies on the generalizability assumption.

**Assumption 1.** A value function of a policy $\pi$ for a specific state-goal pair $(s, g)$ has some generalizability to another state-goal pair $(s', g')$ close to $(s, g)$.

Yet generalizing the policy to unseen state-action pairs requires a large number of samples, given the huge state and action spaces that characterize continuous robotic control problems. As described in Section 2.3, replaying with uniform samples of past experiences without fully exploring pseudo goals leads to an unreliable estimation. To better generalize not just over goals but also states, we preprocess the experiences with a relative goals strategy. Specifically, we denote $g_s$ as the achieved goal and $\tilde{g}$ as the relative goal where $\tilde{g} = g - g_s$. As described in Section 2.1, we obtain the achieved goal from a known and tractable mapping. When it comes to relatives goals space, **Assumption 1** turns to a more common assumption: A value function of a policy $\pi$ for a specific relative goal $\tilde{g}$ has some generalizability to another relative goal $\tilde{g}'$ close to $\tilde{g}$.

In practice, it is more convenient to use relative goals when evaluating the similarity of state-goal pairs, especially when neither states nor goals are close. The strategy provides various samples for generalization over relative goals. After the preprocessing, we manipulate data of the form $(s, \tilde{g})$ to help estimate value function and resample hindsight experiences.

## 3.2 CLUSTERING AND EVALUATION

### VALID EXPERIENCES ACQUIRING

To acquire sufficient data near a state-action pair, we first perform clustering on augmented states in a replay buffer, then estimate the likelihood of a state-action visitation under the current policy.

Firstly, we perform $K$-means clustering over experiences with a distance function, which defines how close of $(s, \tilde{g})$ and $(s', \tilde{g}')$. Specifically, we adopt the Euclidean distance in the definition of $d(*, *)$ since it is task-irrelevant, i.e.

$$d((s, \tilde{g}), (s', \tilde{g}')) = l * d(s, s') + d(\tilde{g}, \tilde{g}'), \tag{3}$$

where the hyperparameter $l$ ensures that the influence of each item won't be omitted. It partitions augmented states into $K$ clusters in which each state belongs to the cluster with the nearest mean. It is easy to sample similar states for a new state after finding the nearest clustering center. Formally, for each $(s, \tilde{g})$, there is $\{(s^i, \tilde{g}^i, a^i) | d((s, \tilde{g}), (s^i, \tilde{g}^i)) < \epsilon_1, i = 1, 2, \ldots, n\}$ where $\epsilon_1$ is small.

Secondly, we estimate the likelihood of a state-action visitation with a batch of similar states. If a state-action pair is well-explored by a policy, the policy will give reliable and stable value estimates for similar states. Hence we define that the likelihood $F(s, \tilde{g}, \pi(s, \tilde{g}))$ is inversely proportional to the max estimate difference, i.e.

$$F(s, \tilde{g}, \pi(s, \tilde{g})) \propto \frac{1}{\max_{1 \leq i \leq n} ||Q^\pi(s, \tilde{g}, \pi(s, \tilde{g})) - Q^\pi(s^i, \tilde{g}^i, \pi(s^i, \tilde{g}^i))||_2}. \tag{4}$$

Thus the estimated value is more reliable for the state-action pair with a high likelihood score, and vice versa. (In practice, it takes desired goal $g$ instead of $\tilde{g}$ as input, i.e. $Q^\pi(s, g, \pi(s, g))$.)

EVALUATION

Approximating the value of a state-action pair from scratch can be difficult with the sparse rewards. Besides evaluating the likelihood of a state-action visitation, we modify the Eq.(2) to give the value a bound related to similar states.

Assume that we have $(s, g) \sim \tau$, $(s', g') \sim \tau'$ and $d((s, g), (s', g')) < \epsilon_1$, where $d(*, *)$ defined in Eq.(3) evaluats the similarity between them and $\epsilon_1 > 0$ is small. Inspired by **Assumption 1**, $(s, g)$ and $(s', g')$ are similar, and as excuting the identical $\pi$, the value function can generalize from one to another. Formally, the generalizability can be specifically expressed as the Lipschitz continuity of state-action value function Ren et al. (2019):

$$|Q^\pi(s, \tilde{g}, \pi(s, \tilde{g})) - Q^\pi(s', \tilde{g}', \pi(s', \tilde{g}'))| \leq Ld((s, g), (s', \tilde{g}')), \quad (5)$$

where $L > 0$ is the Lipschitz coefficient and $d(*, *)$ defined in Eq.(3) evaluats the similarity between $(s, g)$ and $(s', \tilde{g}')$. In the open-ended environment, it's reasonable to claim that the bound Eq.(5) holds for most of the state-goal pairs when $d((s, \tilde{g}), (s', \tilde{g}'))$ is small.

As described in Section 2.3, unsufficient visitations lead to an unreliable estimate. When the likelihood of a state-action visitation is low, it is alternative to sample an action from the excuted ones at the similar state. Therefore, we take the likelihood into account to sample the action $a$ at the next state for the critic. After acquiring a batch of experiences with similar states for the $(s_{t+1}, \tilde{g})$ pair, $\{(s^i, \tilde{g}^i, a^i)|d((s, \tilde{g}), (s^i, \tilde{g}^i)) < \epsilon_1, i = 1, 2, \ldots, n\}$, we modify the approximated value $Q^\pi(s_{t+1}, \tilde{g}, \pi(s_{t+1}, \tilde{g}))$ in Eq.(2) to

$$\underline{Q}^\pi(s_{t+1}, \tilde{g}, a) = \begin{cases} Q^\pi(s_{t+1}, \tilde{g}, \pi(s_{t+1}, \tilde{g})), & F(s_{t+1}, \tilde{g}, \pi(s_{t+1}, \tilde{g})) > \epsilon_2 \\ \max_{1 \leq i \leq n} Q^\pi(s^i, \tilde{g}^i, a^i) - Ld((s_{t+1}, \tilde{g}), (s^i, \tilde{g}^i)), & otherwise \end{cases}$$

where $\epsilon_2 > 0$ is the threshold of the likelihood. It gives a reliable bound of $Q$ value when the action generated by the policy is unfamiliar. Consequently, we perform the value update in Eq.(2) via

$$Q^\pi(s_t, \tilde{g}, a_t) = \mathbb{E}_{s_{t+1}} \left[ r_t + \gamma \underline{Q}^\pi(s_{t+1}, \tilde{g}, a) \right]. \quad (6)$$

**Theorem 1** *Consider the vector space $\mathcal{V}$ over state-goal pairs. For any $U, V \in \mathcal{V}$, we define the metric as $||U - V||_\infty = \max_{(s, \tilde{g}) \in \mathcal{S} \times \mathcal{G}} |U(s, \tilde{g}) - V(s, \tilde{g})|$. Based on Eq.(6), we define the Bellman expectation backup operator $T^\pi$ for policy $\pi$. For each $(s_t, \tilde{g})$,*

$$T^\pi(V(s_t, \tilde{g})) = \mathbb{E}_{a_t, s_{t+1}} \left[ r_t + \gamma \underline{V}^\pi(s_{t+1}, \tilde{g}) \right], \quad (7)$$

*where*

$$\underline{V}^\pi(s_{t+1}, \tilde{g}) = \mathbb{E}_a \left[ \underline{Q}^\pi(s_{t+1}, \tilde{g}, a) \right]$$

*The value iteration formulates as performing $T^\pi$ on vector space $\mathcal{V}$. Then the Bellman expectation backup operator $T^\pi$ with the modified value function is a $\gamma-$ contraction, i.e.*

$$||T^\pi(U) - T^\pi(V)||_\infty \leq \gamma ||U - V||_\infty.$$

The proof is trivial. There is no doubt that it gives the value a reliable bound as well as maintains the convergence of the value update, for that the $T^\pi$ is a contraction.

## 3.3 GOALS REPLAY

We have given a brief description of how HER and its variants sample additional goals from past experiences for replay in Section 1. In short, it is common to prioritize experiences that lead to valuable goals, whilst each method evaluates goals against a different set of criteria. For our method (in Section 3.2), we propose to give a reliable estimate with similar states, which are generated by clustering past experiences. It is feasible only if samples are widely distributed, especially for relative goals. Thus, it is crucial to maintain diverse relative goals in the experiences.

In Robotic Control problems, $\mathcal{G}$ is a set of goals with finite volume. After the preprocessing, each $g \in \mathcal{G}$ leads to various relative goals. Maintaining a broad distribution of relative goals turns to mathematically maximize the entropy of the distribution. Since each relative goal is feasible, maximizing the entropy of the distribution may seem to maintain uniform distribution. Inspired by the skew-fit strategy in (Pong et al., 2019) that iteratively increases the entropy of a generative model, we propose a heuristic operator to manipulate a balanced distribution over valid relative goals.

CURRICULUM GENERATION

Let $\{p_i, i = 1, 2, \dots\}$ be the distributions of relative goals to resample the hindsight experiences. At $i$−th iteration, we sample $N$ augmented states from valid experiences with $p_i$. Since we have partitioned $M$ states into clusters $\{C_k, k = 1, 2, \dots, K\}$, it is trivial to construct empirical distributions of clusters for past valid experiences and sampled hindsight experiences. Formally, by classifying a state to the nearest cluster, we get

$$p_{ve_i}(C_k) = \frac{1}{M} \sum_1^M 1\{(s, \tilde{g}) \in C_k\}, \forall (s, \tilde{g}),$$

$$p_{he_i}(C_k) = \frac{1}{N} \sum_1^N 1\{(s, \tilde{g}) \in C_k\}, (s, \tilde{g}) \sim p_i,$$

where $p_{ve_i}$ for valid experiences and $p_{he_i}$ for hindsight experiences.

Then we firstly skew the $p_{ve_i}$ with the $p_{he_i}$ via

$$p_{skewed_i}(C_k) = \frac{1}{Z_\alpha} p_{ve_i}(C_k) p_{he_i}^\alpha(C_k), \alpha \in [-1, 0],$$

where $Z_\alpha$ is the normalization factor. Secondly, we estimate the Gaussian densities of clusters by calculating the mean $\mu_k$ and the covariance $\sigma_k^2$ of goals for each cluster. We expand $p_{skewed_i}$ to a distribution of goals and fit $p_{i+1}$ to it according to

$$p_{i+1} \leftarrow \sum_k p_{skewed_i}(C_k) \mathcal{N}(\tilde{g}|\mu_k, \sigma_k^2). \tag{8}$$

Repeat the process until we maintain uniform distribution over clusters or a broad distribution of relative goals. The Gaussian mixture model can be trained reasonably fast for RL agents and we take a lazy update on the clustering.

## 4   MULTI-GOAL EXPERIENCE REPLAY

The core idea behind Hindsight experience is that after experiencing some trajectory $\tau$, it stores in the replay buffer every transition $s_t \rightarrow s_{t+1}$ not only with the original goal used for this episode but also with a subset of other goals. To give a reliable estimate for unseen goals, we make use of hindsight experiences as well as try to ensure the generalization over state-action pairs. In particular, we present to resample hindsight experiences with a relative goals strategy (in Section 3). It samples goals from experiences with a high likelihood of the corresponding state-action pair under the current policy and maintains a broad overall distribution of the relative goals.

Here we incorporate our method-Hindsight Curriculum Generation-with vanilla HER, aiming at accelerating the learning in multi-goal tasks with sparse binary rewards. During the process, the updated distributions $\{p_i, i = 1, 2, \dots\}$ automatically form a curriculum as the range of sampled goals progressively expand. Gradually changed probability also strikes a balance between exploitation and exploration.

The detailed algorithm is shown in Algorithm 1, namely Multi-Goal Experience Replay. (For simplicity, we hide the relative goals strategy.)

## 5   EXPERIENMENTS

We employ environments for multi-goal RL introduced in Plappert et al. (2018) and some manipulation tasks are shown in Figure 1. This section is organized as follows. First, we compare the performance between vanilla HER, HER with Energy-Based Prioritization, Curriculum-guided HER and our methods. All the variants are implanted with the DDPG algorithm. Policies are represented as Multi-Layer Perceptrons (MLPs) with Rectified Linear Unit (ReLU) activation functions as in Andrychowicz et al. (2017). Then we construct ablation experiments on major hyper-parameters. The whole training procedure is performed in the simulation. For improved efficiency, we use 20 workers each with 2 rollouts which average the parameters after every update.

---

**Algorithm 1** Multi-Goal Experience Replay

---

**Require:**
- an off-policy RL algorithm $\mathbb{A}$
- a distribution $p$ for resampling goals for replay
- a reward function $r : \mathcal{S} \times \mathcal{A} \times \mathcal{G} \to \mathbb{R}$

Initialize $\mathbb{A}$
Initialize Replay buffer $R$
Initialize $K, l, L, \gamma$
**for** $episode = 1 \to M$ **do**
    Sample a goal $g$ and an initial state $s_0$
    **for** $t = 0 \to T - 1$ **do**
        Sample an action $a_t$ using the behavioral policy from $\mathbb{A}$:
            $a_t \leftarrow \pi_b(s_t \| g)$    ($\|$ denotes concatenation)
        Execute the action $a_t$ and observe a new state $s_{t+1}$
        $r_t := r(s_t, a_t, g)$
        Store the transition $(s_t \| g, a_t, r_t, s_{t+1} \| g)$ in $R$
    **end for**
    Perform the $K$-means clustering in $R$ as describe in Section 3.2
    **for** $t = 0 \to T - 1$ **do**
        Sample a set of additional goal for replay $G$ with $p$ in $R$
        **for** $g' \in G$ **do**
            $r' := r(s_t, a_t, g')$
            Store the transition $(s_t \| g', a_t, r', s_{t+1} \| g')$ in $R$
        **end for**
    **end for**
    **for** $t = 1 \to N$ **do**
        Sample a minibatch $B$ from the replay buffer $R$
        Perform one step of optimization using $\mathbb{A}$ and minibatch $B$ with the modified value function via Eq.(6)
    **end for**
    Update $p$ with via Eq.(8) in Section 3.3
**end for**

---

If the agent achieved the desired goal in an episode, we consider it a success. We show the learning cures of the median success rate of tasks in different settings. The results averages across 5 random seeds and are with shaded areas represent one standard deviation. Experiments demonstrate that our method learns efficiently across different episodes and is robust on the major hyperparameters.

LEARNING CURVE IN MULTI-GOAL SETTING

For all Experience Replay we store each transition in the replay buffer twice: once with the goal used for the generation of the episode and once with the goal corresponding to the future state sampled from the episode. The major difference between the variants is how to sample replay goals, which we state in Section 1 and 3. Notice that there is a modified value iteration for our variant only. All the variant shares the same environment setting and experimental configuration, except for their private controlling parameters. The policies and parameters will be listed in the Appendix.

From the learning curve in Figure 2, we can see the success rates during test rollouts increase with the training process. Our method outperforms all the variant at an early stage, which demonstrates that it improves the learning efficiency. It also maintain a stable high success rate , which is comparable with the vanilla HER algorithm. The learning efficiency indicates that the modified value function matches well with the distribution of tasks. More results will be shown in the Appendix.

ABLATION STUDIES

In this section, we discuss key design choices for our Hindsight Curriculum Generation method that provide substantially improved performance. The major hyperparameter is the Lipschitz coefficient

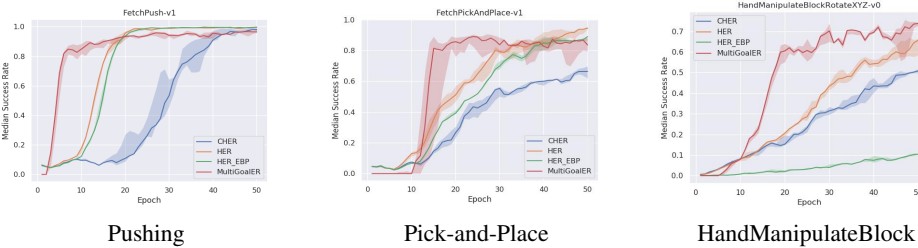

Figure 2: Learning curves for variants with multiple goals. All curves shown are trained with default DDPG hyper-parameters. It shows that at an early stage, our method (labeled MultiGoalER) learns more efficiently and can maintain a stable high success rate.

$L$ that controls the value generalization. Notice that the number of clustering points $K$ is also essential. But we find that the performance is not sensitive when it varies. Thus we keep it constant during all the training episodes. Results in Figure 3 indicates that the choice of $L$ is robust. More results will be shown in the Appendix.

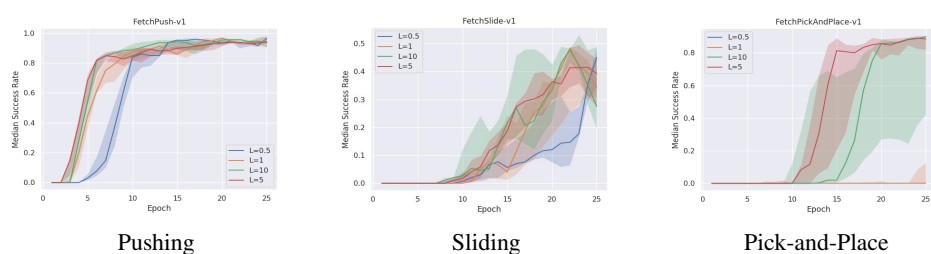

Figure 3: Ablation studies on hyper-parameter $L$. All curves shown are trained with default DDPG hyper-parameters.

## 6    RELATED WORKS

**Hindsights in RL:** HER introduces a hindsight relabelling scheme to extract more information from failures. Temporal Difference Model(TDM) (Pong et al., 2018) generalizes policy to not only unseen goals but also a multi-step temporal scale by relabelling. Hindsight Policy Gradient (HPG)(Rauber et al., 2017) adopts the potential for goal-conditional policies to enable higher-level planning based on subgoals in policy gradient methods. Generalized Hindsight (GH)(Li et al., 2020) converts the data generated from the policy under one task to a different task. Morever, Exploration via Hindsight Goal Generation (HGG)(Ren et al., 2019) constructs a curriculum on goals guiding the exploration of the environment. Dynamic HER (DHER)(Fang et al., 2019a) assembles successful experiences from two relevant failures with dynamic goals.

## 7    CONCLUSION

In this paper, we propose Hindsight Curriculum Generation (HCG) to sample replay experiences that better generalize to unfamiliar goals. It analyzes the likelihood of the corresponding state-action pair under the current policy and the overall distribution of the relative goals. Combing with an off-policy RL trainer, we introduce a multi-goal experience replay method that automatically generates a replay curriculum to progressively expand the range of experiences for training. We have evaluated the method on several Robotic Control problems in the multi-goal tasks suffering from sparse rewards, and experiments demonstrate that it outperforms the state-of-the-art methods on the sample efficiency.

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
