# OpenReview forum: "Hindsight Curriculum Generation Based Multi-Goal Experience Replay"
_ICLR.cc/2021/Conference — Reject_

### Official Review · AnonReviewer4 · 2020-10-28
**Hard to follow; insufficient validation; not enough detail to reimplement; should revise/improve before resubmission**

**Rating:** 3
**Confidence:** 4

**Review:**

The authors introduce a wealth of changes to the standard HER agent and obtain a performance improvement in 3 multi-goal tasks. The main observation made by the paper is that relabeled experiences may be very off-policy / out-of-distribution, and so value estimates for such experiences will be bad.

To help with this the authors propose the following:
- apply K-means to cluster real (state, goal) tuples.
- estimate the likelihood of a given (state, goal) tuple X by finding the closest cluster center in real experience, sampling n real experiences Y^i in that cluster, and taking the minimum of 1/d(X, Y^i).
- they change the Bellman targets in case the likelihood estimate is low. In particular they change it to a lower bound based on some nearby real experience minus the distance times Lipschitz constant.
- they use relative goals (g_original - g_current) instead of absolute goals (g_original).
- there is some kind of curriculum on goal relabeling

This paper is hard to follow. The word usage and sentence structure is unnatural, and I find myself guessing at what exactly the authors mean. This carries through to the math. I think I understand what the modified Bellman backup above equation (6) is doing, but I'm still not entirely sure. I'm also not really following the Section that includes equation (8). As a result, the contributions are a bit unclear.

Theorem 1 is not trivial and there is indeed doubt in my mind. A proof should be provided upon revision.

The related works section can be greatly improved. You should be relating the related work to your own.

An appendix was not provided, despite being referenced, and so the paper is missing additional results (the 3 environments are insufficient), implementation details, and hyperparameter details. Without these, this paper cannot be reimplemented and is not in a publishable state.

Nits:
- Isn't Equation (5) just the definition of Lipschitz continuity, so I'm confused by what is meant by "it's reasonable to claim that [it] holds".
-

---

### Official Review · AnonReviewer1 · 2020-10-29
**An okay paper with many limitations**

**Rating:** 4
**Confidence:** 4

**Review:**

### Summary
This paper focuses on the problem of goal conditioned reinforcement learning. The authors propose an alternative way of performing Bellman updates for goal conditioned value function. Specifically, the proposed method first partitions the space of state goal pairs by performing a K-means clustering, and then estimates the visitation frequency of a state goal pair to be inversely proportional to the maximum difference of Q values within the cluster. For state goal pairs with low visitation frequency, the authors construct a lower bound of the Bellman target value by finding the Q value of a near state goal pair and subtracting the Lipchitz value multiplied by the distance. The authors then use this lower bound as target value to update the Q function. To generate goals for hindsight replay, the authors adopted a skew-fit type method to the empirical distribution of the K-means clusters to sample goals for replay.

The authors evaluate the proposed method on 3 simulated robotics manipulation tasks and compare against HER-EBP and CHER as baselines. The results indicate that the proposed method outperforms prior methods in terms of sample efficiency. The authors also provide ablation studies for the Lipschitz constant hyperparameter.


### Comments
The paper is well written and the idea proposed in this paper is easy to follow. The authors clearly demonstrate the advantage of the proposed method in the 3 robotic manipulation tasks. However, I do have some concerns about the proposed method and results.

First of all, the proposed method seems to be based on some heuristics which have neither been proven to be correct nor been verified empirically. For example, the authors estimate the visitation frequency of a state goal pair to be inverse of max Q function variation within the cluster (equation 4). It is not clear why this is a good estimate, since it is possible for an entire cluster to have low visitation frequency and also low Q value variations because the Q function has not been trained much in the region of the cluster. It would be important to provide either a proof or an empirical study.

Moreover, the proposed method relies on some assumptions that might not hold true for many goal conditioned environments. For example, one assumption lies in the use of L2 distance in equation 3. In many goal conditioned tasks such as maze navigation, the L2 distance might not be a good metric between state goal pairs since two states close in L2 distance could be on two sides of a wall. The paper does not include discussion or empirical evaluations for such tasks.

Thirdly, it is not clear to me the Bellman iteration in equation 6 would converge to the optimal value. The authors only prove that it is a contraction and therefore will converge to some fixed point. However it is unclear to me whether the fixed point would be the same optimal Q value as the unmodified Bellman iteration.

Finally, the proposed method introduces a few extra hyperparameters, such as the cluster K, the Lipchitz constraint of Q function L and the visitation frequency threshold. From the ablation study, we know that the proposed method is sensitive to the choice of L. Therefore, the natural question is whether the observed performance improvement is due to the tuning of these extra hyperparameters. Therefore, it would be important to perform more experiments on a wider range of tasks such as those in [1] and [2].

Due to these limitations, I would not recommend acceptance for this paper before they are addressed.



References

[1] Pong, Vitchyr, et al. "Temporal difference models: Model-free deep rl for model-based control." arXiv preprint arXiv:1802.09081 (2018).
[2] Eysenbach, Ben, Russ R. Salakhutdinov, and Sergey Levine. "Search on the replay buffer: Bridging planning and reinforcement learning." Advances in Neural Information Processing Systems. 2019.

---

### Official Review · AnonReviewer2 · 2020-10-29
**Detailed ablation studies are needed to understand why the approach works; The idea needs to be better motivated and presented.**

**Rating:** 4
**Confidence:** 4

**Review:**

The paper introduces an extension of Hindsight Experience Replay (HER) called Hindsight Curriculum Generation (HCG) which is demonstrated to learn faster in multi-goal RL benchmarks.

The approach consists of two distinct contributions: First, is the idea of learning Q-values as a function of relative goals, instead of absolute goals. Second, is the idea of constructing a distribution over the relative goals from the replay buffer using a simple clustering algorithm and defining a sampling distribution over them.

This approach is then evaluated on three multi-goal RL environments that were previously open-sourced by OpenAI. Learning curves from these domains show that their approach HCG can produce faster learning when compared to the baselines.

Pros:
The approach seems to produce faster learning compared to the baselines considered.

Cons:
The paper seems to be rushed, making it hard to follow the ideas presented in the paper. It also has a few typos.

The motivation for the introduced approach is hard to understand, which makes it difficult to understand why/when the approach works for a given domain. The goal-sampling strategy seems to be arbitrary and no motivation is presented here to justify such an approach.

The related work section in the paper is not detailed. There are many approaches (listed below) that look at discovering curriculums for improving the speed of learning, and these can be considered to be orthogonal to HER, making it applicable to the current setup the authors have considered. The authors need to discuss how their approach of producing a curriculum relates/differs with the curriculum-based approaches.

The algorithm section in the main text does not have the necessary details to help understand the approach. In the pseudocode, many notations are used to present the approach, but I do not see the definitions for them in the main text.

From the experiments, it is not possible to tell whether the improvement in observed performance is due to the idea of using relative goals or the goal-sampling strategy. I would suggest introducing a baseline HER agent that operates on the relative goals, similar to the HCG agent. This should help inform which part of the idea is important.

Theorem 1 in the paper does not seem relevant to the approach and seems to be arbitrarily presented. It would be better if the authors could clarify how this theorem connects to their approach.

Forestier, S., Portelas, R., Mollard, Y., & Oudeyer, P. Y. (2017). Intrinsically motivated goal exploration processes with automatic curriculum learning. arXiv preprint arXiv:1708.02190.

Veeriah, V., Oh, J., & Singh, S. (2018). Many-goals reinforcement learning. arXiv preprint arXiv:1806.09605.

Florensa, C., Held, D., Geng, X., & Abbeel, P. (2018, July). Automatic goal generation for reinforcement learning agents. In International conference on machine learning (pp. 1515-1528).

Graves, A., Bellemare, M. G., Menick, J., Munos, R., & Kavukcuoglu, K. (2017). Automated curriculum learning for neural networks. arXiv preprint arXiv:1704.03003.

---

### Official Review · AnonReviewer3 · 2020-10-30
**The paper has some major issues**

**Rating:** 3
**Confidence:** 4

**Review:**

This paper developed methods for resampling from the hindsight experience replay buffer. The resampling strategy was developed based on the current policy, and the overall distribution of the relative goals. As the distribution over goals evolves over time, the multi-goal agent's replay curriculum is adjusted throughout the learning process. The developed approach, called hindsight curriculum generation (HCG), was applied to DDPG, and evaluated using a set of four robot control problems. Results show that HCG performed better than a few baseline methods, and its performance was claimed to be insensitive to the choice of hyper-parameters.

The paper has a few issues. The sampling from hindsight experience is partially based on the likelihood of the corresponding state-action pair under the current policy. The reviewer is not sure that this strategy makes sense when the developed approach was applied to off-policy RL methods (DDPG in this case). It seems to be suggesting that, without exploration (completely following current policy), off-policy RL methods get the best results. Using only recent policies limits the variance over the collected experience. Some more discussions and justifications are needed for "the likelihood of the corresponding state-action pair under the current policy."

The two baselines of CHER and HER-EBP were not mentioned in the experiment section. The reviewer had to search the whole paper, and found the acronyms mentioned in the introduction section. The results are suspicious: how come CHER and EBP performed even worse than naive HER in Figure 2? The results are inconsistent to those reported in the CHER and EBP papers.

It's stated that "Results in Figure 3 indicates that the choice of L is robust." This is apparently not the case from Figure 3. In the pick-and-place task, when L=5, it reached 0.8 success rate in 15 epochs, whereas the agent couldn't succeed at all when L=1 or L=0.5. The conclusion was not supported by evidence or experimental results.

The paper mentioned "Appendix" in a few places, but there is no appendix in this submission.

It's suggested to experiment with RL methods other than DDPG. There's the potential of applying the developed approach to on-policy methods that have been evident to performing better than DDPG in robot control tasks.

---

### Decision · Program_Chairs · 2021-01-07
**Final Decision**

**Decision:**

Reject

**Comment:**

This paper extends the idea of hindsight experience replay (HER) to learn Q functions with relative goals by constructing a distribution over relative goals sampled from a replay buffer using a clustering algorithm. This approach is evaluated on three multi-goal RL environments and is shown to learn faster than baselines.

${\bf Pros}$:
1. Faster convergence as compared to baselines
2. Interesting use of clustering in the context of HER but this choice is made without strong justifications or formal arguments

${\bf Cons}$:
1. Some of the key choices made in this paper are not justified or explained property, e.g. - the goal sampling strategy, choices made in the clustering algorithm and associated heuristics, implicit assumptions (e.g. R1 raised the question of using L2 distance in measuring metrics between two states)
2. There are several choices made without sufficient formal arguments, verification or guarantees.

The paper studies an interesting problem but could be made stronger by incorporating feedback received during the discussion period.